# Improving Formation Conditions and Properties of *h*BN Nanosheets Through BaF_2_-assisted Polymer Derived Ceramics (PDCs) Technique

**DOI:** 10.3390/nano10030443

**Published:** 2020-02-29

**Authors:** Boitumelo J. Matsoso, Victor Vuillet-a-Ciles, Laurence Bois, Bérangère Toury, Catherine Journet

**Affiliations:** Laboratoire des Multimatériaux et Interfaces, UMR CNRS 5615, Univ Lyon, Université Claude Bernard Lyon 1, F-69622 Villeurbanne CEDEX, France; boijo.matsoso@gmail.com (B.J.M.); laurence.bois@univ-lyon1.fr (L.B.); berangere.toury@univ-lyon1.fr (B.T.)

**Keywords:** *h*BN, PDCs, Li_3_N, borazine, BaF_2_, 2D material

## Abstract

Hexagonal boron nitrite (*h*BN) is an attractive material for many applications such as in electronics as a complement to graphene, in anti-oxidation coatings, light emitters, etc. However, the synthesis of high-quality *h*BN at cost-effective conditions is still a great challenge. Thus, this work reports on the synthesis of large-area and crystalline *h*BN nanosheets via the modified polymer derived ceramics (PDCs) process. The addition of both the BaF_2_ and Li_3_N, as melting-point reduction and crystallization agents, respectively, led to the production of *h*BN powders with excellent physicochemical properties at relatively low temperatures and atmospheric pressure conditions. For instance, XRD, Raman, and XPS data revealed improved crystallinity and quality at a decreased formation temperature of 1200 °C upon the addition of 5 wt% of BaF_2_. Moreover, morphological determination illustrated the formation of multi-layered nanocrystalline and well-defined shaped hBN powders with crystal sizes of 2.74–8.41 ± 0.71 µm in diameter. Despite the compromised thermal stability, as shown by the ease of oxidation at high temperatures, this work paves way for the production of large-scale and high-quality *h*BN crystals at a relatively low temperature and atmospheric pressure conditions.

## 1. Introduction

For over a decade, an enormous scientific research effort has been devoted to the synthesis, tuning, and investigating of various properties and applications of metallic, semiconducting, and insulating two-dimensional (2D) materials. This tremendous research interest in 2D materials is due to the successful isolation of graphene from highly-oriented pyrolytic graphite (HOPG) [1,2,3]. Amongst the most studied 2D materials, hexagonal boron nitride (*h*BN) continues to attract attention due to its unique physicochemical properties. Owing to the strong covalent sp^2^ bonds in the BN plane, *h*BN exhibits a large bandgap (~5.9 eV), high mechanical strength, good thermal conductivity, chemical inertness, and thermal stability. Moreover, the atomically smooth surface and close in-plane lattice mismatch to graphene (~1.8%) [4,5], renders *h*BN an important layered material complementary to graphene and other 2D materials [4,6,7,8,9,10]. As a result, this has led to a myriad of potential applications for *h*BN; ranging from encapsulation of graphene [4,11], tunneling barrier [12], deep ultraviolet light emitters [13], protective coatings and/or lubricants [14,15], hydrogen storage [16], all the way to dielectric substrates [5]. For instance, graphene-based transport devices integrated with an *h*BN dielectric layer have been found to exhibit enhanced mobility and excellent current on-off ratios, compared to those fabricated from graphene stacked on other substrates [5,17,18,19,20,21]. Furthermore, the absence of dangling bonds or trapped charges in *h*BN is of significant importance for enhancing the film performance especially when *h*BN is integrated with transition metal dichalcogenides such as MoS_2_ [22,23,24]. Last but not least, the electrical insulating behavior of *h*BN enables it to serve as a platform for charge fluctuation, contact resistance, gate dielectric, a passivation layer, Coulomb drag, as well as the atomic tunneling layer in a variety of fundamental scientific and technological fields [12,15,23,25,26].

However, for *h*BN to reach its ultimate practical application in the optoelectronic and dielectric industry, synthesis of large-area high-quality single crystals of *h*BN, at cost effective conditions, is a very crucial issue. Consequently, numerous techniques have been developed and employed to produce *h*BN; these include processes such as mechanical exfoliation [5,27,28], sputtering [29,30], pulsed laser deposition (PLD) [31,32], atomic layer deposition (ALD) [33,34,35], and chemical vapor deposition (CVD) [36,37]. For instance, with graphene, mechanical exfoliation has been widely used to produce high-quality *h*BN nanosheets. On the contrary, mechanical exfoliation is strictly limited to the fabrication of small-scale devices as the process produces flakes of a limited size, inconsistent yields, and a variable number of *h*BN layers. Additionally, other techniques basically require extremely sophisticated equipment using high temperatures and pressures [13]. Therefore, in an attempt to circumvent these drawbacks, the use of the polymer derived ceramics (PDCs) [36,37,38,39,40] process coupled with the addition of a crystallization agent such as lithium nitride (Li_3_N) has been reported to provide an alternative approach for preparing large-scale and high-quality single crystals of *h*BN nanosheets for further applications in next-generation electronics. During the PDCs method, a polymeric precursor is synthesized from its monomers, after which it is then converted into ceramic after shaping. Among the various precursors that have been used for the PDCs-synthesis of *h*BN nanosheets, polyborazylene (PBN) has shown to lead to the production of large-area highly crystallized *h*BN at temperatures as low as ~1400 °C [36,37,38,39,40]. This is owing to its relatively high ceramic yield and high purity. Most importantly, the B/N ratio within PBN polymer is ideal to produce stoichiometric *h*BN, with the only contaminants being hydrogen atoms that are easily removed during the ceramization step. Despite the synthesis of large-area and well-crystallized *h*BN nanosheets via the conventional PDCs process, the search for further improvements so as to synthesize well-crystallized *h*BN nanosheets at relatively lower sintering temperatures, still remains a paramount necessity and occupies the activity of today’s scientific community. One such technology, which has been overlooked for almost a decade, is the use of halides in combination with lithium nitride (Li_3_N) [41]. In this process, the Li_3_N acts as a crystallization promoter to produce highly crystallized *h*BN, whereas the halides help to facilitate the melting of Li_3_N, leading to the synthesis of *h*BN nanosheets at temperatures as low as ~800 °C at a prolonged time of 22–56 h. As such, this work reports on the use of PDCs process coupled with Li_3_N-additives in combination with the group II halides, in particular, barium fluoride (BaF_2_), to produce large-area well-crystallized *h*BN nanosheets at low temperatures. The work provides a new platform for the large-scale synthesis of *h*BN nanosheets at cost-effective conditions without compromising the physicochemical properties of *h*BN nanosheets. 

## 2. Materials and Methods 

### 2.1. Procedure 

The pure monomer of borazine was prepared from a reaction between ammonium sulfate ((NH_4_)_2_SO_4_, ≥ 99%, Aldrich, Saint-Louis, MO, USA) and sodium borohydride (NaBH_4_, 98% purity, Aldrich, Saint-Louis, MO, USA) in tetraethylene glycol dimethyl ether or tetraglyme (C_10_H_22_O_5_, ≥ 99%, Alfa Aesar, Ward Hill, MASS, USA) solvent, as reported by Wideman et al [42]. After the purification of borazine via distillation, the polymeric precursor was obtained by through the poly-condensation of borazine at 55 °C inside a pressure-sealed system under argon for 5 d; generating colorless polyborazylene (PBN) [15,39,43,44]. For the synthesis of boron nitrides nanosheets (BNNSs); inside the glove-box and under argon atmosphere, lithium nitride (Li_3_N, 99.4%, Alfa Aesar, Ward Hill, MASS, USA) at a 5 wt.% ratio, as a crystallization agent, and varying amounts (0–10 wt.%) of barium fluoride (BaF_2_, 99%, Alfa Aesar, Karlsruhe, Germany), as a melting-point reduction agent, were added to PBN, then the mixture was homogenized via stirring for 10 min. After which the suspension was heated to 200 °C in an alumina crucible and kept for 1 h to give a solid-state polymer [41]. Finally, the stabilized mixture was annealed for 1 h at 1200 °C (1 °C/min) under inert nitrogen (N_2_, 98%, Air Liquide, Paris, France) atmosphere. 

### 2.2. Methods

The morphology and electronic structure of the synthesized materials were ascertained using various characterization techniques. The crystal structure of the *h*BN nanosheets was confirmed using a powder X-ray diffractometer (Bruker, Billerica, MASS, USA) (PXRD) Bruker D8 Advanced, equipped with the Cu-*K* radiation source and using the PMMA zero-background substrate. MEB Zeiss Merlin Compact scanning electron microscopy (SEM), at the accelerating voltage of 80 kV, was used to determine the morphology of the nanomaterials. Further morphological analyses were carried on the MET Phillips CM120 transmission electron microscope (TEM) (Philips, Amsterdam, The Netherlands) at 120 kV. The degree of crystallization of the *h*BN nanosheets was determined in the backscattering geometry using the HORIBA Jobin-Yvon Labram Evolution Raman spectrometer (Horiba, Kyoto, Japan) at 532 nm laser excitation wavelength. The functional groups and surface interactions of the BN nanosheets nanostructures were investigated using SAFAS Monaco SP2000-IR700 spectrometer in the range of 4000–600 cm^−1^. The surface area, pore volumes, and diameters of the as-prepared samples were acquired from the BELsorpII mini, after degassing the samples for 4 h at 100 °C, thereafter, adsorption and desorption of ultra-pure N_2_ gas was performed. Chemical composition and bonding configurations for the bulk *h*BN samples were determined by XPS using a PHI Quantera SXM spectrometer (Physical Electronics, Chanhassen, MN, USA). A monochromatized aluminum Kα radiation was used with a 200 µm spot diameter and a take-off angle of 45°, before and after 2 µm sputtering with Ar^+^ ions. Charge compensation was provided by an in-lens electron flood gun and separate low energy argon ion source. Finally, thermogravimetric analyses were determined using the TGA/DSC2 form Mettler Toledo (Mettler Toledo, Columbus, Ohio, USA). An *h*BN sample of ~15 µL mass was placed into a 150 mg alumina crucible with pierced lid, after which the decomposition profile of the sample was established from 30–1400 °C, at a heating rate of 20 °C/min, under 50 µL mL^−1^ Ar.

## 3. Results and Discussion

Growth of large-area hBN nanosheets at low temperatures and atmospheric pressure conditions was successfully achieved through a BaF_2_-assisted PDCs technique. After synthesis, the structural, composition and electronic properties were investigated using XRD, XPS, Raman, and IR spectroscopies, whereas the morphological properties were determined using TEM and SEM microscopies.

### 3.1. Structural and Electronic Properties

#### 3.1.1. Powder XRD Analysis

The composition and crystallinity of the BNNS samples synthesized with addition of various amounts of BaF_2_ and annealed at 1200 °C were examined by the powder XRD (Figure 1a), after which their XRD patterns were compared with that of standard *h*BN (ICCD card #: 34–421). The results for all samples showed the characteristic fingerprint diffraction patterns for highly crystallized hexagonal boron nitride (*h*BN) (Figure 1a); evident by the pronounced (002) diffraction peak at 2*θ* ≈ 26.5–26.8°, as well as the less intense (004), (110) and (112) peaks centered at 2*θ* ≈ 55.4°, 2*θ* ≈ 76.2°, 2θ ≈ 81.7°, respectively. However, a much deeper investigation of the diffraction patterns revealed the formation of two phases of hexagonal (*h*BN) and rhombohedral (*r*BN) boron nitride (Figure 1b) for samples with low contents of BaF_2_ (i.e., 0 and 2.5 wt%). On the other hand, upon increasing the BaF_2_ content from 5–10 wt%, the XRD patterns indicated that the most favored principal phase is the *h*BN (Figure 1b), as the relative intensities of the (101) and (012) peaks for *r*BN are relatively diminished. As such, the results showed the significance of the addition of BaF_2_ to the PBN/Li_3_N pre-ceramization mixture towards improving the crystallinity of the *h*BN phase. It can, therefore, be proposed that improved crystallinity of the *h*BN nanosheets is achieved through the facilitation of faster melting of Li_3_N by BaF_2_ [41], thereby leading to lower formation temperature as compared to our previous studies [38,40,43]. However, it is also noteworthy to mention that regardless of the improved crystallinity with increasing BaF_2_ content (10 wt%), the presence of unmelted impurities from Li and Ba complexes is evident from the XRD patterns, thereby compromising the quality of the inherent as-synthesized BNNS samples. This was also observed with a further increase in BaF_2_ content (20–30 wt%, not shown here). Thus, the addition of equally small amounts (5 wt%) of both Li_3_N and BaF_2_ led to the formation of crystalline *h*BN nanosheets at relatively low temperatures and atmospheric conditions. 

A further indication of an improved crystallinity of the *h*BN phase within the BNNS samples was through the separation of the (100) and (101) peaks accompanied by the symmetrically sharpening of (002), as well as increasing intensities of (102), (110), and (112) peaks. Furthermore, the interlayer d_002_-spacing values of ~3.34 Å for all samples were determined to be close to that of high-crystalline bulk value for the commercial *h*BN samples (Table 1), suggesting a good crystallization rate for each sample after the addition of BaF_2_. The slightly lower interlayer distance for the 5 wt% BaF_2_ sample (~3.331 Å) is indicative of the improved d-p interaction occurring between the p-orbital electrons in *h*BN and those in the d-orbital of barium [45], consequently resulting in a better-crystallized sample upon the introduction of BaF_2_. Similar results were observed with the addition of higher BaF_2_ contents, with the 20 wt% BaF_2_ sample registering a d_002_-spacing value of ~3.328 Å, whereas that of the 30 wt% BaF_2_ sample was ~3.326 Å. Finally, the degree of crystallization of the *h*BN phase within the BNNS samples was evaluated in terms of the "graphitization index*, GI*" (Table 1), as indicated by Equation (1):(1)$$GI=\frac{\left[{\left(100\right)}_{area}+{\left(101\right)}_{area}\right]}{{\left(102\right)}_{area}}$$

The significance of the value for the *G.I.* is that the higher the value is the less the three-dimensional ordering is within the *h*BN and the reverse is true [46,47,48,49,50], therefore implying a lateral growth of the *h*BN crystallites. So, in the light of this information, Table 1 shows that BNNS samples synthesized with the addition of 5 wt.% BaF_2_ can be considered to have a less three-dimensional ordering in the crystal structure, as evident by the G.I. value of 3.83c.a. This has later been confirmed by TEM analyses which showed the formation of larger *h*BN nanosheets upon addition of 5 wt% BaF_2_. 

#### 3.1.2. XPS Analysis

The XPS analysis was used as a surface-sensitive and standard technique for determining the overall elemental composition (at. %) and different bonding states within the as-synthesized samples. Appendix A shows a typical surface XPS survey scan for the *h*BN samples and the spectra exhibited at least five peaks: two distinct peaks corresponding to B1s (190.9 eV) and N1s (397.8 eV), two weak peaks corresponding to O1s (532.1 eV) and Ba3d (780.1 eV), as well as another peak corresponding to advantageous C1s (284.2 eV). However, after sputtering a 2 µm surface from the samples with Ar^+^ ions, no carbon was found within the bulk of the samples: thus, indicating the removal of any adsorbed atmospheric carbon atoms. Therefore, by taking the integrated peak areas of the B1s, N1s, O1s, Li1s, and Ba3d from the XPS survey spectra, the overall elemental composition of the bulk samples was determined as a function of increasing BaF_2_, as depicted in Table 2. It can be seen that both boron and nitrogen concentrations increased with addition of more BaF_2_, thereby leading to B/N ratios of 1.62%, 1.64%, 1.57%, and 1.50% for the *h*BN nanosheets grown after addition of 0, 2.5, 5, and 10 wt% BaF_2_, respectively (Table 2). The observed decrease in the B/N ratio corroborated XRD results by indicating that the addition of BaF_2_ makes coalescence of bigger *h*BN domains more, thus compromising the quality of the resultant *h*BN nanosheets. On the other hand, the increasing lithium content between the addition of 0 and 2.5 wt% BaF_2_ is suggestive of the formation of lithium complexes due to the presence of Ba as well as the formation of smaller *h*BN domains, as later confirmed by high-resolution XPS and TGA analysis. Improved growth of *h*BN domains through faster melting of these lithium complexes in the presence of Ba atoms was indicated by the decreasing Li1s content after the addition of 5 and 10 wt% BaF_2_, which was also followed by further reduction of the oxygen content with increasing BaF_2_ content.

To determine the different bonding configurations of each constituent element (B, N, O, Li and Ba) in the bulk *h*BN samples as a function of BaF_2_ content, their high-resolution XPS spectra were fitted with Lorentzian–Gaussian (GL30) peaks using the CasaXPS software. The B1s for all samples exhibited a broad spectral peak with a full width at half maximum (FWHM) was in the range of 1.8–2.7 eV (Figure 2a). This is wider than the reported FWHM value for B in high-quality *h*BN (0.92 eV) [51]. This signifies the presence of different bonding states for B atoms. Therefore, to determine the chemical environments of the B atoms in the nanosheets, the B1s spectral peaks were deconvoluted into three component peaks centered at 190.9–191.0, 192.0–192.1, and 193.2–193.4 eV, respectively. These corresponded to the contribution from sp^2^-BN bonds in high-quality *h*BN, B–O bonds in B_2_O_2_, as well as Li_2_B_4_O_7_ bonds, respectively [51,52,53]. The relatively high intensity for the sp^2^-BN peak component is an indication that the boron atoms are expectedly and predominantly bonded to nitrogen atoms into a hexagonal lattice to form large domains of *h*BN. Furthermore, an increase in the relative concentration of the sp^2^-BN up to addition of 5 wt.% BaF_2_ could be suggestive of the presence of larger *h*BN domains, whilst a decrease in its concentration is an indication of a degradation of the *h*BN domains, but formation of smaller fragments of *h*BN that are prone to be terminated by oxygen atoms. The results thus corroborated XRD results, as the quality of the *h*BN nanosheets was observed to be compromised with the increasing addition of BaF_2_. The formation of larger and better quality *h*BN domains was further ascertained by the position of the B-O bonds as a function of BaF_2_ content. For instance, the positions of the B–O bonds were found to be red-shifted with increasing BaF_2_ content up to the addition of 5 wt.% after which it continued to blue-shift with addition of more BaF_2_. This, therefore, suggested the existence of different bonding states around/within the B–O domains. In particular, the B–O peak position in 0 wt.% BaF_2_
*h*BN nanosheet sample (~193.2 eV) was closer to that of B–O bonds in B_2_O_2_ (~192.55 ± 0.05 eV) [52,54,55], an indication that boron atoms at the defective edges are surrounded by oxygen atoms to form regions of B_2_O_2_ domains. However, on increasing the BaF_2_ content up to addition of 5 wt.%, the B–O peak position red-shifted to ~192.1 eV, an indication that there is substitution of the oxygen atoms in the B_2_O_2_ domains with boron and/or nitrogen atoms due to the formation of larger and better quality *h*BN domains, thereby leading to presence of less saturated B_2_O_2_ domains. Due to the degradation of the structure as a result of impurities from Ba and Li, the peak position of B–O bonds was observed to blue-shift to ~193.4 eV, signifying the formation of B_2_O_2_ domains at the defect regions of the *h*BN domains and/or the small fragments of uncoalescenced PBN.

Confirmation of the bonding states obtained from the B1s was supported by the deconvolution of the XPS N1s spectra (Figure 2b). At least two component peaks centered at 397.9–398.0 eV and 398.9–399.5 eV were observed for all and these were ascribed to the formation of sp^2^N–B and N-H_2_ bonding configurations [36,51,53,56,57]. Lack of the component peak at higher binding energies (i.e., ~401 eV), corresponding to NO_x_ bonding states, is indicative that all the nitrogen atoms within the samples have a high affinity of boron atoms to form *h*BN domains. Furthermore, the bonding states of the oxygen, lithium, and barium atoms was determined by peak fitting of the O1s, Li1s, and the Ba3d high-resolution XPS spectra, as shown in Appendix A. From Appendix A, it can be seen that the O1s was fitted to at least two component peaks. The peaks centered at 531.8–532.4 eV and 529.5–530.1 eV can be ascribed to the contribution from O–B and O–Ba bonds, respectively [54,58]. Interestingly, the component peak at 529.5–530.1 eV could be attributed to the presence of metallic oxide bonds such as O–Li [59,60]; thus, highlighting the ambiguity of using the O1s to assign the bonding configurations of the as-synthesized samples. However, the relative concentrations of the component peaks indicate that most of the oxygen atoms are bonded to the metallic impurities. Moreover, the intensity of the O1s spectra for all samples is observed to decrease with increasing BaF_2_ content, which is in good agreement with the survey spectra, thereby indicating the improved quality with BaF_2_ content. The bonding states of oxygen atoms were confirmed by the deconvolution of the Ba3d spectra (Appendix A), which depicted the presence of *α* (779.8–781.3 eV) and *β* (795.1–798.8 eV) couplings of barium oxide [61,62]. Lack of component peaks corresponding to bonding states such as Ba-*N* [63] is suggestive that the Ba atoms remain bonded to oxygen atoms to form the stable oxide form outside the *h*BN domains, thereby, having no impact on the overall structure of *h*BN nanosheets. Finally, the contribution from the residual lithium was indicated by the peak at ~55.7 eV on the Li1s spectra (Appendix A), which can be attributed to the presence of Li_2_B_4_O_7_ bonds [59,60]. The presence of these bonds can be ascribed to the role played by the crystallization agent (Li_3_N) when it breaks down PBN, to form individual nucleates of *B*-*N*, that can aggregate and grow into *h*BN domains. The relatively large intensity of Li_2_B_4_O_7_ peak at lower BaF_2_ content (0 & 2.5 wt.%) could suggest incomplete decomposition of PBN by Li_3_N, whereas the decreasing intensity of peaks with increasing BaF_2_ content is indicative of the effect of BaF_2_ in dissolving the Li-complexes (i.e., Li_2_B_4_O_7_, Li_2_B_2_O_4_, or Li_3_BN_2_), thereby freeing and permitting boron and/or nitrogen atoms to contribute in the growth of *h*BN domains.

#### 3.1.3. Raman and FT-IR Analysis

Further structural and electronic properties of the as-synthesized BNNS materials were evaluated using Raman spectroscopy. Like graphene and despite the minor difference in the stacking sequence (i.e., AB_graphene_ versus AA′*_h_*_BN_), Raman spectroscopy is also a powerful technique for determining the crystallinity and quality of *h*BN nanomaterials. As a result, Figure 3a display Raman spectra taken from ~10 different areas of the BNNS samples. The spectra showed the first-order active Raman vibrating mode of *h*BN (*E*_2g_) [27,64,65,66,67] centered at ~1365.4 ± 1.6 cm^−1^. To determine the crystallinity of *h*BN materials, studies by Nemanich et al. have reported that there is a direct correlation between the finite-size effects within *h*BN with the inherent position and broadening of the Raman vibrational modes [67]. Their report indicated that the *E*_2g_ vibrational mode blue-shifted and broadened with decreasing crystallites sizes. From Appendix A, the full width at half maximum (FWHM) values for the samples was found to decrease with BaF_2_ content up to 5 wt%, from 17.01 cm^−1^ for the 0 wt% BaF_2_
*h*BN sample to 11.07 cm^−1^ for the 5 wt% BaF_2_
*h*BN sample (Appendix A). The FWHM values thus signify the formation of larger crystallites and subsequently improvement in the crystallinity and quality of the 5 wt% BaF_2_
*h*BN sample. Further confirmation of improved crystallinity was observed through red-shift in the position of *E*_2g_ vibrational mode from ~1366.9 ± 0.25 cm^−1^ for the 0 wt% BaF_2_ sample to ~1365.3 ± 0.21 cm^−1^ for the 5 wt% BaF_2_ sample, after which the peak position blue-shift up to ~1366.4 ± 0.08 cm^−1^ upon increasing the BaF_2_ content to 10 wt.%. The proposed growth mechanism is illustrated in Scheme 1. In summary, the Raman data demonstrated that it is possible to achieve high-quality and crystalline BNNS at atmospheric conditions and reasonably moderate temperatures, in comparison with those *h*BN nanosheets that have been prepared at high temperature and high pressure [13,38,39,40]. 

Fourier transform infrared (FT-IR) spectroscopy constitutes one of the most used techniques for the identification and characterization of phases in BN nanomaterials. This is due to the fact that the sp^2^ and sp^3^ hybridization states of B-N bonds can be easily distinguished by the well-defined adsorption bands [68,69,70]. Therefore, to determine the influence of the addition of BaF_2_ on the surface functionalizations of the *h*BN samples, the FTIR measurements were performed at room temperature. As expected, by two IR active transverse optical (TO) phonon modes of sp^2^ bonded B–N were observed (Figure 3b). The broad and asymmetrical E_1u_ adsorption band at ~1340–1360 cm^−1^ corresponded to the in-plane boron-nitrogen-boron (B–N–B) stretching vibrational modes within one basal plane, whereas the sharp and symmetrical A_2u_ adsorption band, centered at ~740–760 cm^−1^, can be ascribed to the out-of-plane B–N–B bending vibrational modes between two or more basal planes [68,70]. Further analysis of the FT-IR spectra of the modified *h*BN samples revealed a red-shift of the *E*_1u_ adsorption band, whilst the A_2u_ adsorption band was observed to blue-shift with increasing BaF_2_ content.

A slight red shift of the B–N–B stretching mode up to the addition of 5 wt% BaF_2_ relative to the 0 wt.% BaF_2_ sample (Appendix A, blue line-circles) could be suggestive of the addition of strain in the crystal lattice due to the lateral growth of the crystal sizes within the basal plane. However, an introduction of impurities and incomplete and/or growth of individual small crystallites with increasing BaF_2_ content (10–30 wt.%), leads to a blue-shift in the A_2u_ vibrational mode due to removal strain. Similarly, a blue-shift was observed for the E_1u_ mode for 2.5 and 10 wt.% BaF_2_
*h*BN samples in comparison to the 5 wt.% BaF_2_ sample (Appendix A, red line-squares). This can be attributed to the progressive loss of long-range order, in the form of the bond-angle and bond-length disorder, due to the formation of structural defects and distortion of the crystalline structures [71]. The results are in good agreement with the XRD data that indicated the presence of barium and lithium impurities upon the addition of more BaF_2_. On the contrary, a slight red-shift for the 5 wt.% BaF_2_
*h*BN sample further supported the formation of large-area *h*BN nanosheets, thus leading to the addition of strain in between the basal plane.

#### 3.1.4. Thermal Stability Investigation

The thermogravimetric analysis (TGA) curves depicting the normalized percentage mass-change of the as-synthesized *h*BN samples as a function of temperature are illustrated in Figure 3c. The plots showed that the decomposition of the samples can be considered to occur in at least four steps. The samples are observed to be stable up to ~60 °C, after which subsequent decomposition begins at different on-set temperatures (region I), which can be ascribed to the varying crystallinity of the samples. In particular, the on-set temperature for the 0 wt% BaF_2_ sample was determined to be ~74 °C; whilst those for modified samples were found to be ~67 °C, ~82 °C, and ~69 °C, for 2.5, 5, and 10 wt% BaF_2_
*h*BN samples, respectively. The highest on-set temperature for the 5 wt% BaF_2_ sample is in corroboration with XRD and Raman analyses which indicated improved crystallinity, thus indicating the difficulty of breaking the stable bonds at relatively low temperatures. On the other hand, the lowest onset decomposition temperature for 2.5 wt% BaF_2_ sample can be attributed to the incomplete restructuring of the *h*BN lattice to form larger crystallites due to insufficient amount of BaF_2_. Owing to the formation of these incomplete crystallites, the entire bond structure is weakened, thereby leading to faster degradation. In the case of 10 wt% BaF_2_ sample, the earlier onset decomposition temperature can be ascribed to the disruption of the lattice of the *h*BN structure due to the fast melting of Li_3_N in the presence of larger content of BaF_2_ and formation of impurity complexes, as depicted by XRD analysis. Consequently, a weakened structure that is prone to easy loss of hydrogen atoms and faster decomposition is formed. Region II, with mass losses occurring at temperatures between 150–600 °C, is due to the decomposition of lower energy bonds, such as dehydrogenation of intercalated H_2_ (~436 kJ/mol) between the *h*BN nanosheets as well as hydrogen atoms bonded to the edge-defects (i.e., B–H bonds at ~330 kJ/mol; N-H bond at 314 kJ/mol; O–H bond at ~428 kJ/mol; adsorbed H–OH bonds at ~498 kJ/mol) [72,73,74] within the *h*BN lattice. The decomposition of the entire *h*BN lattice also occurs within this region (i.e., B–N bond at ~398 kJ/mol). Decomposition and removal of other compounds such as Ba–F (~487 kJ/mol), Ba–OH (~477 kJ/mol), N-F (~301 kJ/mol), Li-H (~247 kJ/mol), Li–OH (~427 kJ/mol), and/or Li–F (~577 kJ/mol) [72,73,74] are also observed within this region. The decomposition of these low energy bonds is thus depicted by the presence of two peaks on the DTG curves (Appendix A). Between 600–1200 °C (region III), an increase in mass of 0.22%, 0.35%, and 0.42% was observed 2.5, 5, and 10 wt% BaF_2_
*h*BN samples, respectively. On the contrary, further decrease in the decomposition profile of the 0 wt% BaF_2_
*h*BN sample was recorded in this region, and corresponds to the removal of α-Li_3_BN_2_ and β-Li_3_BN_2_ complexes at ~860 °C and 920 °C, respectively [75,76], depicted by XPS. Thereafter, the final degradation of the 0 wt% BaF_2_
*h*BN sample is observed beyond 1200 °C. Interestingly, the 0 wt% BaF_2_
*h*BN sample does not decompose completely to 0 wt% (Appendix A), however, only ~1.2 mg (8.52%) of the sample was decomposed (region IV). This can be attributed to the oxidation of BN by residual O_2_ inside the TGA oven from the inert gas used (i.e., Ar, 99%, Alphagaz), leading to the subsequent formation of thin layer of thermally stable boron trioxide (B_2_O_3_, ~1850 °C_boiling_, and ~1500 °C_sublimation_) [77,78] on the surface of the nanosheets as well as evolution of a certain amount of nitrogen oxides (NO*_x_*) [79], which ultimately prevents further decomposition. Upon addition of 2.5 wt% BaF_2_, a mass increase of ~0.22% (~ 30.1 µg) in region III occurred within a temperature range of ~910–1220 °C, attributed to the removal of the α-Li_3_BN_2_ complexes as well as the formation of thermally stable B_2_O_3_ layer. However, with the addition of more BaF_2_ from 5 to 10 wt%, the mass increase was recorded to be ~0.27% (~36.0 µg) and ~0.42% (~61.5 µg) in the temperature ranges of ~630–1330 °C and ~770–1310 °C, respectively. In both cases, the faster formation of the thermally stable B_2_O_3_ layer could suggest the lack of the Li_3_BN_2_ complexes as a result of the faster melting of Li_3_N, which was facilitated by the addition of BaF_2_, thereby corroborating the XRD data. The rate of oxidation for the BaF_2_-samples can be ascribed to the purity, crystallinity and specific surface area (SSA) [80,81]. For instance, highly-crystalline BN with a small SSA, as in the case of the 5 wt% BaF_2_
*h*BN nanoplatelets, provides less reactive sites for oxidation and thus a smaller weight gain, in comparison to moderately-crystalline BN with a small SSA (10 wt% BaF_2_
*h*BN) and poorly-crystalline BN (2.5 wt% BaF_2_
*h*BN). Beyond 1350 °C (region IV), the decomposition of the B_2_O_3_ layer by the residual boron atoms leads to the formation and removal of the gaseous dioxodiborane compounds (B_2_O_2_) [82,83,84], as based on Equation (2):(2)23B(g)+23B2O3(l)→B2O2(g)
The differential scanning calorimetry (DSC) curve (Appendix A) provided important information regarding the heat-flow arising by a series of physical or chemical procedures, such as decomposition, oxidation, as a function of temperature. An endothermic peak was observed for 0 and 5 wt.% BaF_2_
*h*BN samples at ~151.1 °C and ~147.1 °C, respectively; whereas no thermal phenomena were observed for other samples (Appendix A). This corresponded to the endothermic reactions associated with the dehydrogenation of the samples. Further endothermic reactions associated with the decomposition of all samples were depicted by the decrease in the heat flow in the temperature range of ~480–620 °C (Appendix A). Moreover, small exothermic peaks appeared in the temperature range of ~640–1400 °C (Appendix A), indicating the oxidation of the samples as shown by the increasing mass on TGA thermograms and hence confirming the transformation of BN to B_2_O_3_. All parameters related to the TGA/DSC measurements are presented in Appendix A.

#### 3.1.5. Surface Area Determination

Surface area and pore-size distribution of *h*BN are essential properties for its potential application in energy storage and conversion devices as well as in biotechnological applications. As such the specific surface area (SSA), pore-size distribution of the as-synthesized *h*BN samples were determined using the multi-point Brunauer–Emmet-Teller (BET) method through adsorption/desorption measurements of N_2_ at 77K. From the N_2_ adsorption/desorption isotherm curves (Figure 4a), the *h*BN synthesized with the addition of BaF_2_ demonstrated a type II isotherm with increasing BaF_2_ content; an indication of the formation of macroporous or non-porous materials [85]. Minor N_2_ amounts were adsorbed under relatively lower relative pressures (*P*/*P*_0_ < 0.01) for all samples, with no hysteresis loop being observed under low pressure (*P*/*P*_0_ < 0.45); an indication of the absence of micropores and mesopores on the materials’ surface. Interestingly, the hysteresis loop was observed to decrease between adsorption/desorption under higher pressures (*P*/*P*_0_ > 0.45) with increasing BaF_2_ content. Thus, the physical adsorption mechanism on the as-synthesized *h*BN samples can be described as follows: without BaF_2_ content (i.e., 0 wt.%), the N_2_ adsorption is a formation of an unrestricted multilayer after the completion of a monolayer, followed by delayed desorption of N_2_. Thus, this accounts for the high specific surface area recorded for the 0 wt.% BaF_2_
*h*BN sample (~8.7 m^2^/g), which can then be attributed to the presence of a relatively small amount of macropores (Figure 4b) and/or the external rough surface [80,86]. On the other hand, with the addition of BaF_2_ content, the adsorption mechanism can be presumed to be following the adsorption and desorption of the monolayer of N_2_ onto the material’s external rough surface of the basal planes due to the non-porous morphology of the samples. This is shown by the adsorption curve which does not reach a plateau at a relatively high pressure close to unity (*P*/*P*_0_ ∼ 0.99), but rather extends indefinitely. Therefore, based on the multi-point BET method, the specific surface areas for the samples were then determined from the BET plots extracted using points between 0.05 < *P*/*P*_0_ < 0.30 (Appendix A) and these were found to be 8.7, 3.5, 3.6, and 2.9 m^2^/g for samples prepared after addition of 0, 2.5, 5, and 10 wt% of BaF_2_, respectively. Appendix A gives a summary of the textual properties of these samples. 

### 3.2. Morphological Analysis

The morphological properties of the *h*BN nanosheets obtained at 1200 °C after the addition of varying amounts of BaF_2_ were investigated by scanning electron microscopy (SEM). Figure 5a displays the typical SEM micrographs of the as-synthesized 5 wt.% BaF_2_
*h*BN nanosheets, whereas the micrographs of other samples, as well as the commercial sample of hBN (99.5%, Alfa Aesar), are represented in Appendix A. The dimensions of crystals of the *h*BN samples were then determined to be 2.74 ± 0.67, 8.41 ± 0.71, and 5.30 ± 0.31 µm for samples annealed after the addition of 2.5, 5, and 10 wt.% of BaF_2_, respectively. On the contrary, it was difficult to determine the flake-size for the 0 wt.% Ba F_2_
*h*BN sample since the sample was an agglomerated mass of irregular shaped and jagged-edged plate-like crystals (Appendix A). However, upon the introduction of BaF_2_, the morphology of the crystals becomes more defined (Figure 5a and Appendix A); like that of the well-defined disc-shaped and homogenous nanoplatelets of the commercial *h*BN sample (Appendix A). For instance, with the addition of 5 wt.% BaF_2_, the *h*BN nanosheets display a much more well-defined, smooth-edged and homogenous plate-like morphology. This is expected as both Raman and XRD data suggested an improved crystallinity of the *h*BN nanosheets upon the addition of 5 wt% BaF_2_. Although an increase in the BaF_2_ content to 10 wt% (Appendix A) also led to the formation of well-defined and plate-like nanosheets, the size of the nanosheets was compromised as evident by the breakage of the nanoplatelets. More degradation in the size of the nanoplatelets was also observed with samples synthesized with the addition of 20 and 30 wt% BaF_2_. The formation of smaller *h*BN nanosheets with increasing BaF_2_ content may be attributed to the abrupt melting of Li_3_N due to the presence of more cations form BaF_2_, thus consequently leading to incomplete crystallization of *h*BN from PBN. Similar results were reported by various groups, whereby they showed that not only does the different usage of cations led to the formation of different morphologies of *h*BN nanosheets, but an increase in these cations resulted in compromised morphologies of such *h*BN nanostructures [87,88]. The final determination of the morphological properties of the *h*BNNS samples was performed using the transmission electron microscopy (TEM). Low magnification TEM micrographs (Figure 5b and Appendix A) illustrated that the samples are mainly composed of overlapping sheet-like structures, with crystal sizes of 0.89 ± 0.01, 2.88 ± 0.74, 3.32 ± 0.25, and 3.15 ± 0.67 µm, for samples annealed after addition of 0, 2.5, 5, and 10 wt.% of Ba F_2_, respectively. Remarkably, the low-magnification TEM images further corroborated the SEM micrographs by depicting the formation of thinner and larger *h*BN nanosheets, with well-defined disc-shape, after the addition of 5 wt% of BaF_2_. 

## 4. Conclusions

Hexagonal boron nitride (*h*BN) nanosheets exhibiting well-defined morphology and large crystal size were successfully synthesized at low temperatures (1200 °C) and atmospheric pressure through modification of the polymer-derived ceramics (PDCs) technique with varying amounts of barium fluoride (BaF_2_). The XRD, Raman, and XPS data revealed the formation of highly-crystalline *h*BN nanosheets with the FWHM_E2g_ of 11.07 cm^−1^ and a G.I. value of *c.a.* 3.83 upon addition of 5 wt% BaF_2_ to the pre-ceramization mixture of PBN and 5 wt% Li_3_N. Morphological analysis revealed formation well-defined shape for the nanosheets with an average size ranging from 2.74 ± 0.67 µm to 8.41 ± 0.71 µm from SEM and 2.88 ± 0.74 µm to 3.32 ± 0.25 µm from TEM after chemical exfoliation. On the other hand, thermogravimetric analysis showed that the addition of BaF_2_ led to the formation of less stable samples, as evidenced by the ease of oxidation at high temperatures. However, this work paves the way for the production of large-scale and good-quality *h*BN crystals at relatively low temperature and atmospheric pressure conditions.

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
