# Peer review of "Improving Formation Conditions and Properties of hBN Nanosheets Through BaF2-assisted Polymer Derived Ceramics (PDCs) Technique"

_nanomaterials, 2020, doi:10.3390/nano10030443_

Round 1

Reviewer 1 Report

In this work, the authors have demonstrated production of h-BN nanosheet at 1200 C. Several characterization techniques have been employed to determine quality of the h-BN material. 

Comments:

(i) Abstract discusses h-BN powders, whereas main text discusses h-BN nanosheet – am I missing something?

(ii)  Sec 3.1.1:  rBN phase is present for 2.5% BaF2 case, but not for 5%BaF2 case! Comments?

Author Response

Please the attachement.

Reviewer 2 Report

In this paper on “Improving formation conditions and properties of hBN nanosheets through BaF2-assisted PDCs technique”, the authors have shown an interesting approach to prepare hexagonal boron nitrite crystals at relatively low temperatures and pressures. However, there are some inconsistencies in the discussion of results, which are given in the below scientific comments. In addition, authors are suggested to improve the presentation of the manuscript, figures and terminologies used in the paper. Therefore, I recommend for the publication of this paper in Nanomaterials only after a major revision.

General comments.

Authors need to pay close attention to the typos, spelling mistakes and missing words. All over the manuscript, terminology of symbols and abbreviations has been randomly used and persuaded. Just to provide some examples, ~5 eV (correct) and ~ 5 eV (incorrect); 0-10 wt% instead of 0 – 10 wt% or 0 - 10 wt%.; ºC versus oC; Fig. or Figure versus figure, and Table versus table, etc. A pdf version of the paper with yellow color background heighted regions is attached to point out the author to more edits.

In the title replace PDCs with ‘polymer derived ceramics’ because non-specialists are not aware of PDCs. Authors terminology in mentioning wt% is totally random in all over the manuscript such as, wt.%, wt%, wt. % etc., Random colors have been used to show the results corresponding to hBN nanosheets obtained by the addition of 0, 2.5, 5 and 10 wt% of BaF2. For example, 5 wt% BaF2 is in cyan (Figure 1) and in blue (Figure 1 inset), where as in Magenta color (Figure 2). I suggest the authors to maintain the consistency of colors across different figures. There are several formatting issues in Figure 1, rBN/hBN or rBN/hBN? 5 wt% instead of 5wt%. Presentation of XPS results in Figures is extremely confusing. Some signals are deconvoluted, but not others. In the main text (Figure 2), top curve is 10 wt.%, and in the SI the bottom curve is shown to be 10 wt%. Manuscript requires a thorough check; Line 23 (paves the way...), L148 (evidence.….. visible?), L126 (BaF2-assisted…), L188 (survey spectra, the overall), L233 (Due to the..) and L396 (samples were determined…)

Scientific comments.

While authors used different characterization techniques to compare the crystallinity and morphology, and bulk properties such as surface area and porosity, the results and discussion section states “After synthesis, the structural, composition and electronic properties were invested ….”. Neither full 3D crystal structures (or local structures using solid-state NMR spectroscopy) nor electronic properties are reported in this paper. Instead, different hBN compounds are characterized using XRD, XPS, Raman and IR spectroscopies. This statement needs to be modified. XPS results show the peaks correspond to binding energies associated with carbon atoms. However, these peaks were not seen after etching/sputtering the surface 2 mm deep with Ar+ In the later part of the paper, SEM and TEM micrographs show existence of crystalline regions between 2.7 and 5 mm. If the nanosheets are carbon contaminated on the surface and up to 2 mm deep from the surface, then significant fraction hBN nanosheets are indeed contaminated (?). In addition, it has been mentioned that the addition of BaF2 leads to less stable BNs and other oxidation products. Authors need to address this by carrying out more accurate characterization of the materials, for example, using solid-state NMR spectroscopy of neat PBN, and 0, 2.5, 5 and 10 wt% BaF2-treated BN compounds. Specifically, the characterization of different aggregates of PBN fragments, rBN and hBN compounds is required. 13C and 11B (also 15N, if possible) solid-state NMR spectroscopy measurements and analyses are expected to provide evidences for the local structures and different 13C and 11B local environments in rBN and hBN compounds. Although the surface area determination using BET curves explain that there is a reduction in the porosity upon the incorporation of BaF2, the molecular-level origins for the reduced surface area in BaF2-treated BN compounds is not clear and undiscussed. It needs to be clarified. To support the arguments in L23 and L469 (production of large-scale and good-quality crystals at relatively low temperature), 13C and 11B solid-state NMR analyses need to carried out and discussed in the paper.

Reviewer 3 Report

The reviewed ”Improving formation conditions and properties of hBN nanosheets through BaF2-assisted PDCs technique” authors covered important issues regarding with synthesis of large-area hBN nanosheets by the modified polymer-derived ceramics (PDCs) technique with varying amounts of barium fluoride (BaF2). The article is a continuation of the topics previously presented by the authors in the articles published in e.g. ACS Appl. Nano Mater., Nanotechnology and Crystals. The issues raised are presented at a high scientific level with great attention to details (both in relation to research methodology as well as used materials and equipment) and adequate graphics (XRD patterns, TEM/SEM micrographs) taken from results of carried out experiments. The article contains a few minor errors/mistakes that must be eliminated. Few of them are listed below:

Page: 1, Line: 15: Acronym PDS (appears first time) should be explained. Page: 1, Line: 21/24: Should be hBN rather than hBN. Page: 1, Line: 37: Please add space between (~1.8 %) and [4]. Page: 1, Line: 38: Please add space between materials and [4]. Page: 2, Line: 52/73: Please eliminate the conjunction hanging at the end of the line – move "a" to the next line. Page: 2, Line: 78: Should be °C rather than °C. Page: 3, Line: 117: Should be 45° rather than 45 °. Page: 3, Line: 136-138: Value and unit should be in the same line. Page: 4, Line: 162: Move ”(~” to the next line. Page: 6, Line: 219: Please eliminate the conjunction hanging at the end of the line – move "a" to the next line. Page: 6, Line: 241: Move ”(i.e. ~” to the next line. Page: 8, Line: 288: Move ”10” to the next line. Page: 8, Line: 310: Move ”0” to the next line. Page: 9, Line: 341: Word ”Region” should be written as a bold. Page: 11, Line: 406: Please eliminate the conjunction hanging at the end of the line – move "a" to the next line. The authors used many symbols and acronyms. I suggest considering inserting their explanation in the Nomenclature at the end of the work (after the Conflicts of Interest).

Round 2

Reviewer 2 Report

It seems authors improved presentation of results in the manuscript. I therefore recommend this paper for a publication in Nanomaterials.